# A Fundamental Accuracy–Robustness Trade-off in Regression and Classification

## Abstract

We derive a fundamental trade-off between standard and adversarial risk in a rather general situation that formalizes the following simple intuition:

> *If no (nearly) optimal predictor is smooth, adversarial robustness comes at the cost of accuracy.*

As a concrete example, we evaluate the derived trade-off in regression with polynomial ridge functions under mild regularity conditions.

## 1 Introduction

The study of *adversarial robustness* of machine learning models is concerned with finding prediction schemes whose accuracy gracefully degrades when minute but adversarial perturbations are applied to the data. We focus on a simple prevalent mathematical abstraction of the problem that can be described as follows. Given a pair of random variables $(X, Y) \in \mathbb{R}^d \times \mathbb{R}^k$ that represent a data sample and its corresponding label, as well as $\mathcal{F}$, a class of functions from $\mathbb{R}^d$ to $\mathbb{R}^k$, the goal is to find an $f \in \mathcal{F}$ that achieves a low adversarial risk $\mathbb{E}\left(\sup_{\Delta \,:\, \|\Delta\| \leq \epsilon} \ell\left(f\left(X + \Delta\right), Y\right)\right)$ where $\ell \colon \mathbb{R}^k \times \mathbb{R}^k \to \mathbb{R}_{\geq 0}$ is the loss function and $\|\cdot\|$ is a certain norm defined over $\mathbb{R}^d$.

Intuitively, if a prediction function $f \in \mathcal{F}$ is very non-smooth (e.g., it has a large *Lipschitz seminorm*), then we can expect its adversarial risk

$$R_\epsilon(f) = \mathbb{E}\left(\sup_{\Delta \,:\, \|\Delta\| \leq \epsilon} \ell\left(f\left(X + \Delta\right), Y\right)\right),$$

to be much larger than its standard risk

$$R(f) = \mathbb{E}\left(\ell\left(f\left(X\right), Y\right)\right).$$

We formalize this intuition and show a simple fundamental trade-off between the standard and adversarial risks of any candidate function $f \in \mathcal{F}$ as quantified by certain notion of local sharpness of $f$. It is worth mentioning that the constraints on the intensity of the data perturbation are formulated using a norm merely for the sake of a simpler exposition; many other types of perturbation constraints can be addressed by straightforward adaptation of the presented arguments.

In the finite sample setting we only access $n$ independent draws of $(X, Y)$ which we denote by $(X_1, Y_1), \ldots, (X_n, Y_n)$. Then, the empirical risk and its adversarial version are defined respectively by

$$\widehat{R}_n(f) = \frac{1}{n} \sum_{i=1}^{n} \ell(f(X_i), Y_i),$$

and

$$\widehat{R}_{n,\epsilon}(f) = \frac{1}{n} \sum_{i=1}^{n} \sup_{\Delta_i \,:\, \|\Delta_i\| \leq \epsilon} \ell(f(X_i + \Delta_i), Y_i).$$

*Empirical risk minimization* (ERM) is the most common mechanism to construct a predictor $f$ from $n$ data samples that "generalizes" well in the sense that $R(f)$ is close to $\min_{\widetilde{f} \in \mathcal{F}} R(\widetilde{f})$. Specifically, based on the premise that $R(\cdot)$ is nowhere much larger than $\widehat{R}_n(\cdot)$, ERM provides the predictor

$$\widehat{f}_n = \operatorname*{argmin}_{f \in \mathcal{F}} \widehat{R}_n(f) \,.$$

Analogously, empirical adversarial risk minimization, also referred to as *adversarial training*, provides the predictor

$$\widehat{f}_{n,\epsilon} = \operatorname*{argmin}_{f \in \mathcal{F}} \widehat{R}_{n,\epsilon}(f) \,. \tag{1}$$

If $\widehat{R}_{n,\epsilon}(f)$ concentrates around $R_\epsilon(f)$ uniformly for all $f \in \mathcal{F}$, which can be shown, e.g., using a variety of tools from the theory of empirical processes (van der Vaart & Wellner, 2012) or PAC-Bayesian arguments (Catoni, 2007), under reasonable regularity conditions, then we can guarantee with high probability that $R_\epsilon(\widehat{f}_{n,\epsilon}) \leq \min_{f \in \mathcal{F}} R_\epsilon(f) + o_n(1)$ where $o_n(1)$ is a term that vanishes to zero typically at rate $n^{-1/2}$. Therefore, the adversarial risk of $\widehat{f}_{n,\epsilon}$ is nearly-optimal and we only need to examine the gap between the standard risk of $\widehat{f}_{n,\epsilon}$ and the optimal standard risk $\min_{f \in \mathcal{F}} R(f)$. We do not pursue the finite sample scenario any further in this paper. Instead, we exclusively focus on a fundamental trade-off between the standard population risk $R(f)$ and its adversarial analog $R_\epsilon(f)$ that exist for any arbitrary predictor $f \in \mathcal{F}$ even if the access to the data distribution is not restricted by finite samples.

**Related Work**

A comprehensive review of the literature on adversarial robustness is beyond the scope of this work, but we summarize some of the results in this area that are most relevant for us. For a broader view of the literature interested readers are referred to (Carlini et al., 2019; Bai et al., 2021) and references therein.

A common theme in the literature is to analyze adversarial robustness in classification or regression problems assuming a curated data distribution (Fawzi et al., 2017; Schmidt et al., 2018; Tsipras et al., 2019; Dan et al., 2020; Javanmard et al., 2020; Javanmard & Soltanolkotabi, 2022; Dobriban et al., 2023; Javanmard & Mehrabi, 2024). Among the results that study the trade-offs between standard and adversarial risk, (Javanmard et al., 2020) considers the least squares linear regression with standard Gaussian covariates under an $\ell_2$ adversarial perturbation. Leveraging the convex Gaussian min-max theorem (CGMT) (Thrampoulidis et al., 2015) they provide a precise (asymptotic) trade-off formula between $R_\epsilon(\widehat{f}_{n,\epsilon})$ and $R(\widehat{f}_{n,\epsilon})$ where $\widehat{f}_{n,\epsilon}$ is the linear function obtained by adversarial training on $n$ samples as in (1). This trade-off formula approaches a fundamental limit for any estimator by increasing the considered "sampling ratio" $n/d$. Similarly, (Javanmard & Soltanolkotabi, 2022) considers binary classification under the Gaussian mixture model and derives a precise formula for the standard, robust classification accuracy, and a phase transition threshold for the sampling ratio at which the classes become "robustly separable". Even though (Javanmard & Soltanolkotabi, 2022) considers the test accuracy in terms of the binary loss function, the classifier is learnt based on a suitable decreasing convex loss function not only to have a closed-form expression for the robust risk, but also to have a setup compatible with the CGMT.

Adversarial $\ell_2$ and $\ell_\infty$ robustness for classification of a mixture of two or three Gaussian distributions with colinear means and identical isotropic covariance matrices is analyzed in (Dobriban et al., 2023). In the mentioned setting, optimal and approximately optimal robust classifiers are derived, and it is shown that the trade-off between accuracy and robustness for any classifier deteriorates as the imbalance of the classes increases.

Binary classification with $\ell_p$-adversaries for two particular low-dimensional manifolds is analyzed in (Javanmard & Mehrabi, 2024) where the covariates are modeled as $X = \varphi(WZ)$ with $\varphi$ denoting a monotonic coordinatewise nonlinearity, $W \in \mathbb{R}^{d \times k}$ being a tall matrix, and $Z \in \mathbb{R}^k$ being the low-dimensional latent variable. The first model is a Gaussian mixture model where $Y$ is a biased $\pm1$-valued random variable, and conditioned on $Y$ we have $Z \sim \text{Normal}(Y\mu, \text{I})$ for a fixed $\mu \in \mathbb{R}^k$. The second model is a *generalized linear*

*model* where $Z \sim \text{Normal}(0, I)$ and $\mathbb{P}(Y = 1 \mid Z) = 1 - \mathbb{P}(Y = -1 \mid Z) = g(\beta^\intercal Z)$ for a monotonic "link function" $g \colon \mathbb{R} \to [0, 1]$ and a fixed $\beta \in \mathbb{R}^k$. With $\sigma_{\min}(W)$ denoting the smallest singular value of $W$, it is shown in (Javanmard & Mehrabi, 2024) that if $\sigma_{\min}(W)$ dominates $\epsilon d^{1/2 - 1/p}$ as $d \to \infty$, then for both of the considered models the gap between the adversarial risk and the standard risk (i.e., the "boundary risk") of the optimal standard classifier vanishes asymptotically.

**Contributions and Remarks**

In contrast with the work mentioned above, our main motivation has been to expose the role of "sharpness" (as a notion opposite to "smoothness"), through an explicit and formal mathematical formulation of the trade-off between the adversarial and standard risks that holds in rather general scenarios under minimal assumptions. To do so, we introduce a mild condition on the loss function to satisfy certain "three-point quasi triangle inequalities" which readily hold for the commonly used loss functions. These simple conditions allow us to derive a lower bound for the sum of the standard and adversarial risk of any function in terms of a certain notion of the sharpness of the function. Suppose that for given data distribution and function class, there are functions that achieve an adversarial risk comparable with the optimal standard risk achieved within the function class. Therefore, based on our result, the sharpness of these functions has to be no more that a multiple of the optimal standard risk. This requirement, imposed on the provided function class and the data distribution determines the tolerable $\epsilon$ that limits the adversary's power.

As a concrete example, we use our approach to derive a trade-off between the standard and adversarial risk in the problem of regression over polynomial ridge functions, which includes linear regression as a special case. We obtain a threshold value for $\epsilon$ beyond which adversarial robustness cannot hold without a significant sacrifice of the regression accuracy. Our result also suggests that the adversarial robustness deteriorates by increasing the nonlinearity as characterized by the degree of the polynomial ridge function.

We emphasize that, while detailed analysis of specific models can be valuable, we must be cautious about generalizing the observed behavior under these models. For example, most of the models for which adversarial robustness is analyzed are regression and (binary) classification models with data distributions for which there are "simple" (e.g., linear or affine) (nearly) optimal predictors. While such models are amenable to highly detailed analysis, we need to consider more general scenarios in order to identify and understand the central mechanisms that create the trade-off between the standard and adversarial risk.

**Future Direction**

The importance of certain isoperimetric properties of the data distribution in adversarial robustness has been observed in prior work (e.g., (Bubeck & Sellke, 2023), (Dobriban et al., 2023)). As can be extracted from the analysis of the regression problem in Section 3, our work exposes the importance of alternative characteristics of the data distribution that can be expressed through a non-standard functional inequality that resembles the classic Poincaré inequality. A deeper understanding of such functional inequalities in a general setting is critical to understanding adversarial robustness.

## 2 Problem Setup and the Main Result

The basic property that we will exploit is that there often exist functions $A : \mathbb{R}^k \times \mathbb{R}^k \to \mathbb{R}_{\geq 0}$ and $B : \mathbb{R}^k \times \mathbb{R}^k \to \mathbb{R}_{\geq 0}$ such that $A(u, u) = B(u, u) = 0$ for all $u \in \mathbb{R}^k$, and for every two pairs $(u, v)$ and $(u', v')$ in $\mathbb{R}^k \times \mathbb{R}^k$ we have

$$\ell(u, v) + \ell(u', v') + A(u, u') \geq B(v, v'), \tag{2}$$

and

$$\ell(u, v) + \ell(u', v') + A(v, v') \geq B(u, u'). \tag{3}$$

For the sake of concreteness, we distinguish three special scenarios in our notation. The first scenario is the *least-squares regression* where $k = 1$ and

$$\ell(u, v) = \ell_{\mathsf{LS}}(u, v) \stackrel{\text{def}}{=} (u - v)^2/2$$

for which, using the Cauchy-Schwarz inequality,

$$A(u, v) = A_{\mathsf{LS}}(u, v) \stackrel{\text{def}}{=} (u - v)^2/2$$

and

$$B(u, v) = B_{\mathsf{LS}}(u, v) \stackrel{\text{def}}{=} (u - v)^2/6$$

fulfill the conditions (2) and (3). The second scenario is the *multiclass classification* where the functions $f \in \mathcal{F}$ map their input to the unit simplex $\triangle^{k-1} \subset \mathbb{R}^k$ for some $k > 1$ and the response variable $Y$ is some extreme point of $\triangle^{k-1}$, i.e., a canonical basis vector in $\mathbb{R}^k$. In this setting, the Kullback–Leibler (KL) divergence can be used as the loss function, i.e.,

$$\ell(u, v) = \ell_{\mathsf{KL}}(u, v) = \sum_{j=1}^{k} v_j \log \frac{v_j}{u_j},$$

for which, by Pinsker's inequality (see, e.g., (Boucheron et al., 2013, Theorem 4.19)), the functions

$$A(u, v) = A_{\mathsf{KL}}(u, v) \stackrel{\text{def}}{=} \frac{1}{2} \|u - v\|_1^2,$$

and

$$B(u, v) = B_{\mathsf{KL}}(u, v) \stackrel{\text{def}}{=} \frac{1}{6} \|u - v\|_1^2,$$

satisfy the conditions (2) and (3).

Another suitable loss function for multiclass classification whose corresponding risk is the actual misclassification probability, rather than a surrogate of it, is

$$\ell(u, v) = \ell_{0/1}(u, v) = \begin{cases} 0, & \text{if } u = v \text{ or } \exists i \in [k] \text{ such that } u_i > u_j \text{ and } v_i > v_j \text{ for all } j \in [k] \backslash \{i\} \\ 1, & \text{otherwise}, \end{cases}$$

with $[m] \stackrel{\text{def}}{=} \{1, 2, \ldots, m\}$ for any positive integer $m$. Clearly, $\ell(u, v) = \ell(v, u)$, and we have

$$\ell_{0/1}(u, v) + \ell_{0/1}(u', v') + \ell_{0/1}(u, u') \geq \ell_{0/1}(v, v'),$$

since the left-hand side is either greater than 1 or the vectors $u$, $u'$, $v$, and $v'$ all have their unique maximum at the same coordinate. Therefore, in this case we can choose

$$A(u, v) = A_{0/1}(u, v) = \ell_{0/1}(u, v),$$

and

$$B(u, v) = B_{0/1}(u, v) = \ell_{0/1}(u, v).$$

**Theorem 1.** *Assuming that the loss function $\ell \colon \mathbb{R}^k \times \mathbb{R}^k \to \mathbb{R}_{\geq 0}$, and the functions $A \colon \mathbb{R}^k \times \mathbb{R}^k \to \mathbb{R}_{\geq 0}$ and $B \colon \mathbb{R}^k \times \mathbb{R}^k \to \mathbb{R}_{\geq 0}$ meet the conditions (2) and (3), then for every $f \in \mathcal{F}$ we have*

$$R(f) + R_\epsilon(f) \geq \sup_{(X', Y') \sim (X, Y)} \max \left\{ \mathbb{E} \left( \sup_{\Delta \colon \|\Delta\| \leq \epsilon} B(f(X), f(X' + \Delta)) - A(Y, Y') \right), \right.$$

$$\left. \mathbb{E} \left( B(Y, Y') - \inf_{\Delta \colon \|\Delta\| \leq \epsilon} A(f(X), f(X' + \Delta)) \right) \right\},$$

*where, the outer supremum is with respect to the pair of random variables $(X', Y')$, which may depend on $(X, Y)$, and has the same joint distribution as $(X, Y)$. In particular, we also have the simplified bound*

$$R(f) + R_\epsilon(f) \geq \max \left\{ \mathbb{E} \left( \sup_{\Delta \,:\, \|\Delta\| \leq \epsilon} B(f(X), f(X + \Delta)) \right), \mathbb{E}\, B(Y, Y') \right\},$$

*in which, conditioned on $X$, $Y'$ and $Y$ are identically distributed.*

*Proof.* Under (3), we can write

$$\ell(f(X), Y) + \ell(f(X' + \Delta), Y') \geq B(f(X), f(X' + \Delta)) - A(Y, Y').$$

Taking the supremum with respect to $\Delta$ subject to $\|\Delta\| \leq \epsilon$, and then taking the expectation on both sides of the inequality yields

$$R(f) + R_\epsilon(f) = \mathbb{E}\, \ell(f(X), Y) + \mathbb{E} \left( \sup_{\Delta \,:\, \|\Delta\| \leq \epsilon} \ell(f(X' + \Delta), Y) \right)$$

$$\geq \mathbb{E} \left( \sup_{\Delta \,:\, \|\Delta\| \leq \epsilon} B(f(X), f(X' + \Delta)) - A(Y, Y') \right). \tag{4}$$

Similarly, it follows from (2) that

$$\ell(f(X), Y) + \ell(f(X + \Delta), Y') \geq B(Y, Y') - A(f(X), f(X' + \Delta)).$$

Again, taking the supremum with respect to $\Delta$ subject to $\|\Delta\| \leq \epsilon$, and then taking the expectation on both sides of the inequality yields

$$R(f) + R_\epsilon(f) = \mathbb{E}\, \ell(f(X), Y) + \mathbb{E} \left( \sup_{\Delta \,:\, \|\Delta\| \leq \epsilon} \ell(f(X' + \Delta), Y') \right)$$

$$\geq \mathbb{E} \left( B(Y, Y') - \inf_{\Delta \,:\, \|\Delta\| \leq \epsilon} A(f(X), f(X' + \Delta)) \right). \tag{5}$$

The claimed lower bound on $R(f) + R_\epsilon(f)$ follows by choosing the better lower bound between (4) and (5).

The simplified bound follows by considering two special choices for $(X', Y')$. Choosing $(X', Y')$ to be identical to $(X, Y)$ reduces the right-hand side of (4) to $\mathbb{E} \left( \sup_{\Delta \,:\, \|\Delta\| \leq \epsilon} B(f(X), f(X + \Delta)) \right)$. If instead we choose $X' = X$, with $Y'$ and $Y$ being independent and identically distributed conditioned on $X$, the right-hand side of (5) reduces to $\mathbb{E}\, B(Y, Y')$. □

It is worth highlighting the role of the two terms on the right-hand side of the simplified bound provided by Theorem 1. The first term, i.e., $\mathbb{E} \left( \sup_{\Delta \,:\, \|\Delta\| \leq \epsilon} B(f(X), f(X + \Delta)) \right)$, becomes significant if $f(\cdot)$ is not (locally) smooth, whereas the second term, i.e., $\mathbb{E}\, B(Y, Y')$, becomes significant if $f(\cdot)$ is (locally) smooth, but the measurement/label noise is large.

The following immediate corollary addresses the cases of multiclass classification and least-squares regression mentioned above where the risk of a function $f \in \mathcal{F}$ is $R(f) = \mathbb{E}\, \ell_{\mathsf{KL}}(f(X), Y)$ and $R(f) = \mathbb{E}\, \ell_{\mathsf{LS}}(f(X), Y)$, respectively.

**Corollary 1.** *Define the "mean local sharpness factor" of $f \in \mathcal{F}$ with respect to $X$ as*

$$L_\epsilon(f) = \mathbb{E} \sup_{\Delta \,:\, \|\Delta\| \leq \epsilon} \|f(X + \Delta) - f(X)\|_1^2,$$

*both in the case of least-squares regression and multiclass classification (with $\ell_{\mathsf{KL}}(\cdot)$ as the loss function). Then, we have*

$$R(f) + R_\epsilon(f) \geq \frac{1}{6} \max \left\{ L_\epsilon(f), \mathbb{E}\, \|Y - Y'\|_1^2 \right\}, \tag{6}$$

*with $Y$ and $Y'$ being i.i.d. conditioned on $X$.*

*Proof.* The claim follows immediately from Theorem 1 by specializing $B(\cdot)$ to $B_{\mathsf{LS}}(\cdot)$ in the case of least squares regression, and to $B_{\mathsf{KL}}(\cdot)$ in the case of multiclass classification. □

Corollary 1 shows that for every $f \in \mathcal{F}$ the standard risk $R(f)$ and the adversarial risk $R_\epsilon(f)$ cannot be both less than $\frac{1}{12} \max \left\{ L_\epsilon(f), \mathbb{E} \|Y - Y'\|_1^2 \right\}$. In particular, because the adversarial risk of a function is always greater than its standard risk, any function $f \in \mathcal{F}$ whose standard risk $R(f)$ is nearly optimal in the sense that[1] $R(f) \lesssim R_\star = \inf_{\widetilde{f} \in \mathcal{F}} R(\widetilde{f})$, cannot achieve an adversarial risk $R_\epsilon(f)$ comparable to $R_\star$ if it is not sufficiently smooth as quantified by $L_\epsilon(f) \gg R_\star$. From a different perspective, the derived trade-off can be interpreted as the necessity of $\epsilon$ to be sufficiently small such that $L_\epsilon(f) \lesssim R_\star$, to make $R(f) + R_\epsilon(f) \lesssim R_\star$ possible.

The following is a similar corollary in the case of multiclass classification using $\ell_{0/1}(\cdot)$ as the loss function.

**Corollary 2.** *For any $\mathcal{S} \subset \mathbb{R}^d$ define the $\epsilon$-core of $\mathcal{S}$ with respect to the norm $\|\cdot\|$ as*

$$\mathrm{core}_\epsilon(\mathcal{S}) \stackrel{\mathrm{def}}{=} \left\{ x \in \mathbb{R}^d \colon \{x\} + \epsilon \mathcal{B} \subseteq \mathcal{S} \right\},$$

*where $\mathcal{B}$ denotes the unit ball of $\|\cdot\|$, and the sum of two sets is taken to be their Minkowski sum. For any $f \colon \mathbb{R}^d \to \mathbb{R}^k$ let*

$$\mathcal{S}_i(f) = \left\{ x \in \mathbb{R}^d \colon f_i(x) > \max_{j \in [k] \setminus \{i\}} f_j(x) \right\},$$

*denote the set of points $x \in \mathbb{R}^d$ at which the ith coordinate of $f(x)$ is the unique largest entry of $f(x)$. Then, for multiclass classification using the $\ell_{0/1}(\cdot)$ loss, we have*

$$R(f) + R_\epsilon(f) \geq \max \left\{ \mathbb{P} \left( X \notin \cup_{i \in [k]} \mathrm{core}_\epsilon \left( \mathcal{S}_i(f) \right) \right), \frac{1}{2} \mathbb{E} \|Y - Y'\|_1 \right\},$$

*where, conditioned on $X$, $Y'$ is an independent copy of $Y$ (whose domain is the set of canonical basis vectors in $\mathbb{R}^k$).*

*Proof.* In view of Theorem 1, it suffices to show that

$$\mathbb{E} \sup_{\Delta \colon \|\Delta\| \leq \epsilon} \ell_{0/1}(f(X), f(X + \Delta)) = \mathbb{P} \left( X \notin \cup_{i \in [k]} \mathrm{core}_\epsilon \left( \mathcal{S}_i(f) \right) \right), \tag{7}$$

and

$$\mathbb{E} \, \ell_{0/1}(Y, Y') = \frac{1}{2} \|Y - Y'\|_1. \tag{8}$$

Recalling the definition of $\ell_{0/1}(\cdot)$, we have $\sup_{\Delta \colon \|\Delta\| \leq \epsilon} \ell_{0/1}(f(x), f(x + \Delta)) = 0$ if for all $\Delta \in \epsilon \mathcal{B}$, the maximum entry of $f(x + \Delta)$ occurs at the same unique coordinate. This can be equivalently expressed as $x \in \cup_{i \in [k]} \mathrm{core}_\epsilon \left( \mathcal{S}_i(f) \right)$, from which (7) follows. Furthermore, (8) holds because the fact that $Y$ and $Y'$ are canonical basis vectors in $\mathbb{R}^k$ guarantees that

$$\ell_{0/1}(Y, Y') = \frac{1}{2} \|Y - Y'\|_1. \qquad \square$$

An intuitive interpretation of Corollary 2 is that if the functions $f \in \mathcal{F}$ that provide near-optimal standard accuracy, in the sense that $R(f) \lesssim R_\star = \inf_{\widetilde{f} \in \mathcal{F}} R(\widetilde{f})$, have small $\epsilon$-cores in regions where each coordinate of $f$ is dominant, then adversarial robustness comes at the cost of losing the accuracy.

Of course, the bounds established in Theorem 1, Corollary 1, and Corollary 2 inevitably contain abstract terms due to the generality of these bounds. However, these abstract terms can be approximated appropriately using the structure of the special prediction problems of interest. In the next section, we consider a special regression problem and express the derived bounds in terms of more explicit quantities.

---

[1]We write $a \lesssim b$ if $a \leq Cb$ for some absolute constant $C > 0$.

# 3 Least Squares Regression over Polynomial Ridge Functions

Consider the case of least squares regression over the set of polynomial ridge functions

$$\mathcal{F}_p = \left\{ x \mapsto f_\theta(x) \stackrel{\text{def}}{=} \langle \theta, x \rangle^p : \theta \in \mathbb{R}^d \right\},$$

for some integer $p \geq 1$. In particular, for a parameter $\theta_\star \in \mathbb{R}^d$ the observations have the form

$$Y = f_{\theta_\star}(X) + Z,$$

where $X$ is a zero-mean random variable in $\mathbb{R}^d$ whose marginals have finite moments of order at least $2p$, and the noise term $Z$ is a random scalar independent of $X$ such that $\mathbb{E} Z = 0$ and $\mathbb{E} Z^2 = \sigma^2$. This model reduces to the standard linear regression model for $p = 1$. With $\Sigma$ denoting the covariance matrix of $X$ and $\|\theta\|_\Sigma \stackrel{\text{def}}{=} (\theta^\mathsf{T} \Sigma \theta)^{1/2}$, we further assume that for some constant $C_p > 0$, for every $\theta \in \mathbb{R}^d$ we have

$$\left( \mathbb{E} \, |\langle \theta, X \rangle|^{2p} \right)^{1/(2p)} \leq C_p \|\theta\|_\Sigma. \tag{9}$$

For $f_\theta(x) = \langle \theta, x \rangle^p \in \mathcal{F}_p$ the quantity $L_\epsilon(f_\theta)$ can be expressed as

$$L_\epsilon(f_\theta) = \mathbb{E} \sup_{\Delta: \, \|\Delta\| \leq \epsilon} |\langle \theta, X + \Delta \rangle^p - \langle \theta, X \rangle^p|^2$$

$$= \mathbb{E} \left( (|\langle \theta, X \rangle| + \|\theta\|_* \epsilon)^p - |\langle \theta, X \rangle|^p \right)^2$$

where the second line follows from the facts that

$$(|z_0| + \delta)^p - |z_0|^p \leq \sup_{z: \, |z - z_0| \leq \delta} |z^p - z_0^p|,$$

as one can specialize the right-hand side to the case $z = z_0 + \text{sgn}(z_0)\delta$, and

$$\sup_{z: \, |z - z_0| \leq \delta} |z^p - z_0^p| = \sup_{z: \, |z - z_0| \leq \delta} \left| \sum_{k=1}^p \binom{p}{k} (z - z_0)^k z_0^{p-k} \right|$$

$$\leq \sum_{k=1}^p \binom{p}{k} \delta^k |z_0|^{p-k}$$

$$= (|z_0| + \delta)^p - |z_0|^p.$$

To obtain a lower bound for $L_\epsilon(f_\theta)$ we will use the following identities

$$\mathbb{E} \left( (|\langle \theta, X \rangle| + \|\theta\|_* \epsilon)^p - |\langle \theta, X \rangle|^p \right)^2 = \mathbb{E} \left( \sum_{k=1}^p \binom{p}{k} |\langle \theta, X \rangle|^{p-k} \|\theta\|_*^k \epsilon^k \right)^2 \tag{10}$$

$$\mathbb{E} \left( \sum_{k=1}^K \binom{p}{k} |\langle \theta, X \rangle|^{p-k} \|\theta\|_*^k \epsilon^k \right)^2 = \sum_{j=1}^K \sum_{k=1}^K \binom{p}{j} \binom{p}{k} \mathbb{E} \, |\langle \theta, X \rangle|^{2p-j-k} \|\theta\|_*^{j+k} \epsilon^{j+k}, \tag{11}$$

for $K \in [p]$. Using the fact that for any pair of nonnegative numbers $a$ and $b$ we have $(a + b)^2 \geq a^2 + b^2$, it follows from (10) and (11) that

$$\mathbb{E} \left( (|\langle \theta, X \rangle| + \|\theta\|_* \epsilon)^p - |\langle \theta, X \rangle|^p \right)^2 \geq \|\theta\|_*^{2p} \epsilon^{2p} + \mathbb{E} \left( \sum_{k=1}^{p-1} \binom{p}{k} |\langle \theta, X \rangle|^{p-k} \|\theta\|_*^k \epsilon^k \right)^2$$

$$= \|\theta\|_*^{2p} \epsilon^{2p} + \sum_{j=1}^{p-1} \sum_{k=1}^{p-1} \binom{p}{j} \binom{p}{k} \mathbb{E} \, |\langle \theta, X \rangle|^{2p-j-k} \|\theta\|_*^{j+k} \epsilon^{j+k}$$

$$\geq \|\theta\|_*^{2p}\epsilon^{2p} + \sum_{j=1}^{p-1}\sum_{k=1}^{p-1}\binom{p}{j}\binom{p}{k}\|\theta\|_\Sigma^{2p-j-k}\|\theta\|_*^{j+k}\epsilon^{j+k}$$

$$= \|\theta\|_*^{2p}\epsilon^{2p} + \left(\sum_{k=1}^{p-1}\binom{p}{k}\|\theta\|_\Sigma^{p-k}\|\theta\|_*^k\epsilon^k\right)^2$$

$$\geq \frac{1}{2}\left(\sum_{k=1}^{p}\binom{p}{k}\|\theta\|_\Sigma^{p-k}\|\theta\|_*^k\epsilon^k\right)^2,$$

where we used the power mean inequality on the third line and the Cauchy–Schwarz inequality on the fifth line. Therefore, we have shown that

$$L_\epsilon(f_\theta) \geq \frac{1}{2}\left((\|\theta\|_\Sigma + \|\theta\|_*\epsilon)^p - \|\theta\|_\Sigma^p\right)^2.$$

Furthermore, we have

$$\mathbb{E}(Y - Y')^2 = 2\sigma^2.$$

Applying the inequalities above in (6), we obtain

$$R(f_\theta) + R_\epsilon(f_\theta) \geq \max\left\{\frac{1}{12}\left((\|\theta\|_\Sigma + \|\theta\|_*\epsilon)^p - \|\theta\|_\Sigma^p\right)^2, \frac{\sigma^2}{3}\right\}.$$

Because the standard risk is of the form

$$R(f_\theta) = \mathbb{E}\left(\langle X, \theta\rangle^p - \langle X, \theta_\star\rangle^p\right)^2 + \sigma^2,$$

if $R(f_\theta) + R_\epsilon(f_\theta) \lesssim R(f_{\theta_\star}) = \sigma^2$, meaning that $f_\theta(X)$ is an accurate and robust predictor for $Y$, then we must have

$$\mathbb{E}\left(\langle X, \theta\rangle^p - \langle X, \theta_\star\rangle^p\right)^2 \lesssim \sigma^2,$$

and

$$\left((\|\theta\|_\Sigma + \|\theta\|_*\epsilon)^p - \|\theta\|_\Sigma^p\right)^2 \lesssim \sigma^2.$$

With

$$\lambda_* = \sup_{\vartheta \neq 0}\frac{\|\vartheta\|_\Sigma^2}{\|\vartheta\|_*^2},$$

we can write

$$\left((\|\theta\|_\Sigma + \|\theta\|_*\epsilon)^p - \|\theta\|_\Sigma^p\right)^2 = \|\theta\|_\Sigma^{2p}\left(\left(1 + \frac{\|\theta\|_*}{\|\theta\|_\Sigma}\epsilon\right)^p - 1\right)^2$$

$$\geq \|\theta\|_\Sigma^{2p}\left(\left(1 + \frac{\epsilon}{\sqrt{\lambda_*}}\right)^p - 1\right)^2.$$

Furthermore, using the triangle inequality and the moments equivalence assumption (9), we also have

$$\sqrt{\mathbb{E}\langle X, \theta_\star\rangle^{2p}} \leq \sqrt{\mathbb{E}\left(\langle X, \theta\rangle^p - \langle X, \theta_\star\rangle^p\right)^2} + \sqrt{\mathbb{E}\langle X, \theta\rangle^{2p}}$$

$$\leq \sqrt{\mathbb{E}\left(\langle X, \theta\rangle^p - \langle X, \theta_\star\rangle^p\right)^2} + C_p^p\|\theta\|_\Sigma^p.$$

Combining the derived bounds we obtain

$$\mathbb{E}\langle X, \theta_\star\rangle^{2p} \lesssim \left(1 + \frac{C_p^p}{(1 + \epsilon/\sqrt{\lambda_*})^p - 1}\right)^2\sigma^2.$$

Interpreting $\mathsf{SNR}_p = \mathbb{E}\,\langle X, \theta_\star \rangle^{2p}/\sigma^2$ as the Signal-to-Noise-Ratio and using the inequality $(1+\epsilon/\sqrt{\lambda_\star})^p - 1 \geq \max\{p\epsilon/\sqrt{\lambda_*}, (\epsilon/\sqrt{\lambda_*})^p\}$, the bound above shows that if

$$\epsilon \gg \min\left\{ \frac{C_p^p}{p}\sqrt{\frac{\lambda_*}{\mathsf{SNR}_p}},\, C_p\sqrt{\frac{\lambda_*}{\mathsf{SNR}_p^{1/p}}} \right\}, \tag{12}$$

adversarial robustness is impossible unless we are operating at low $\mathsf{SNR}_p$ meaning that the optimal standard risk is also relatively large.

In particular, for linear regression which corresponds to the case of $p = 1$, where we have $\mathsf{SNR}_1 = \|\theta_\star\|_\Sigma^2/\sigma^2$ and $C_1 = 1$, adversarial robustness is impossible if

$$\epsilon \gg \sqrt{\frac{\lambda_*}{\|\theta_\star\|_\Sigma^2}}\,\sigma\,,$$

unless $\|\theta_\star\|_\Sigma^2/\sigma^2$ is low. In particular, assuming that $X$ has the identity matrix as its covariance (i.e., $\Sigma = I$) and the adversarial perturbations are bounded in $\ell_2$ norm (i.e., $\|\cdot\| = \|\cdot\|_2$), we have $\lambda_\star = 1$. Therefore, the threshold above reduces to $\epsilon \gg \sigma/\|\theta_\star\|_2$. Up to the constant factors, this threshold is equivalent to what can be calculated using the analytic formula for the adversarial risk (see, e.g., (Javanmard et al., 2020, Lemma 3.1)). Assuming that $X$ has a Gaussian distribution, Javanmard et al. (2020) provide precise asymptotic expressions for the standard and adversarial risk of the adversarially trained estimator, as the dimension and the sample size grow proportionally to infinity.

It is worth mentioning that the threshold for $\epsilon$ specified by (12) is in general dependent on the dimension $d$ through the parameter $\lambda_*$. For example, if $\Sigma = I$ and the perturbations are bounded in $\ell_\infty$ norm (i.e., $\|\cdot\| = \|\cdot\|_\infty$), then we have $\lambda_* = \sup_{\vartheta \neq 0} \|\vartheta\|_2^2/\|\vartheta\|_1^2 = 1/d$. Therefore, if $p$, $C_p$, and $\mathsf{SNR}_p$ are constants independent of the dimension $d$, robustness against adversarial $\ell_\infty$ perturbations of size greater than $O(d^{-1/2})$ cannot be guaranteed.

We can also examine the effect of the nonlinearity in the model, as parameterized by $p$, using (12). For simplicity, let us focus on the special case where $X$ is sub-Gaussian which can be expressed by (9) with $C_p = C_1\sqrt{p}$ for a sufficiently large constant $C_1 > 1$ and all $p \geq 1$. Operating at a fixed $\mathsf{SNR}_p = \eta > C_1^{2p}p^{p-2}$ and $\epsilon \leq \sqrt{\lambda_\star}$, the impossibility threshold (12) simplifies to

$$\epsilon \gg C_1^p p^{p/2-1}\sqrt{\frac{\lambda_\star}{\eta}}\,,$$

which is more restrictive for smaller values of $p \geq 2$.

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
