# OpenReview forum: "A Fundamental Accuracy--Robustness Trade-off in Regression and Classification"
_TMLR — Rejected by TMLR_

### Review · Reviewer_4FXk · 2024-11-18

**Summary Of Contributions:**

This paper investigates in a theoretical manner the accuracy-robustness trade-off for regression and multi-class classification tasks. It provides lower bounds for both standard and adversarial risk in terms of $\epsilon$-perturbation size. Furthermore, building on these derived bounds, the paper presents bounds for a more explicit example of a polynomial ridge function applied to least-squares regression.

**Audience:**

Yes

**Broader Impact Concerns:**

No broader impact concerns

**Claims And Evidence:**

Yes

**Requested Changes:**

This paper addresses an important area of the accuracy-robustness- trade-off. While it indicates under what conditions adversarial robustness can or cannot be guaranteed— e.g. based on the dimensions of the data for rigid polynomial regression in the context of $\ell_{\infty}$  adversarial attacks—the above open questions remain.

**Strengths And Weaknesses:**

**Strengths**:
- This topic is important and studied in a broader context compared to the existing literature.
- This paper offers insights into perturbation size and their influence on the adversarial robustness in terms of lower bounds.

**Weaknesses**:
- A connection to other mentioned existing studies ([1]-[5]), examining this trade-off, would help assess the contributions of this paper.
- While this paper investigates a relatively general scenario, it remains unclear how this can be translated to the statements from the other mentioned papers.
- What are the immediate follow-up steps? How can we utilize these findings? A discussion or outlook would help here.
- On page 5, the statement ("From a different perspective, the derived trade-off can be interpreted as the necessity of $\epsilon$ to be sufficiently small such that $L_{\epsilon}(f) \underset{\sim}{<} R^*$, to make $R(f) + R_{\epsilon}(f)  \underset{\sim}{<}  R^*$ possible.") results in the fact that a small perturbation size is necessary for achieving a nearly optimal trade-off between standard risk and adversarial risk. However, this point seems rather obvious.

Overall, I feel that this paper offers a valuable theoretical examination of an existing intuition, improving over already existing theoretical analyses which use specific assumptions. This study provides lower bounds for the perturbation size, which would still allow for adversarial robustness, however does not sufficiently demonstrate the influence of this finding to commonly used adversarial attacks under $\ell_2$ and $\ell_{\infty}$ perturbations in e.g., image classification scenarios. Providing further information on this aspect and its practical applications would enhance the significance of this study.  Thus, I would appreciate some small experimental analysis.



[1] Muni Sreenivas Pydi, Varun Jog. The Many Faces of Adversarial Risk. NeurIPS, 2021

[2]Adel Javanmard, Mohammad Mehrabi. Adversarial robustness for latent models: Revisiting the robust-standard accuracies trade-off. Operations Research, 2024

[3] Christos Thrampoulidis, Samet Oymak, Babak Hassibi. Regularized linear regression: A precise analysis of the estimation error. COLT, 2015

[4] Adel Javanmard, Mahdi Soltanolkotabi, Hamed Hassani. Precise tradeoffs in adversarial training for linear regression.COLT, 2020

[5] Edgar Dobriban, Hamed Hassani, David Hong, Alexander Robey. Provable trade-offs in adversarially robust classification. IEEE Transactions on Information Theory, 2023

---

> ### Author Response · Authors · 2025-02-12
>
> We thank the reviewer for his/her comments. Below is our point-by-point response.
>
> > A connection to other mentioned existing studies ([1]-[5]), examining this trade-off, would help assess the contributions of this paper.
>
> Reference [1] is mainly concerned about measurability issues that arise in various definitions of adversarial risk and the mathematically rigorous ways to fix these issues. In the problems of interest for us the source of any potential measurability issue is taking supremum of a stochastic process over a ball. As is common practice in probability theory, in these cases we can define the supremum as the limit of the supremum over increasingly finer epsilon-nets of the ball, which avoids the measurability issues because epsilon-nets are finite sets.
>
> Reference [3] develops a mathematical tool and is not specifically about adversarial robustness. The results of [2], [4], and [5] are of course more detailed, but they are focused on very special models. Our goal is to develop a framework that can be applied more generally. Of course, for any special problem, the thresholds derived through a general approach might not be as precise as those tailored to the specifics of the special problem. Towards the end of Section 3 we have expanded our discussion in the case of linear regression (i.e., p = 1) and made comparison with the results of [4].
>
> > While this paper investigates a relatively general scenario, it remains unclear how this can be translated to the statements from the other mentioned papers.
>
> In principle, our results can be applied in those special classification and regression problems similar to the regression example provided in Section 3. In fact, the analysis of Section 3 can be generalized to derive a threshold for ϵ beyond which adversarial risk is significantly larger than the optimal standard risk. The critical factor in this threshold is a quantity that resembles the Poincaré constant of the data distribution.
>
> > What are the immediate follow-up steps? How can we utilize these findings? A discussion or outlook would help here.
>
> We have added a subsection for Future Direction at the end of the Introduction.
>
> > On page 5, the statement ("From a different perspective, the derived trade-off can be interpreted as the necessity of ϵ to be sufficiently small such that Lϵ(f)<∼R∗, to make R(f)+Rϵ(f)<∼R∗ possible.") results in the fact that a small perturbation size is necessary for achieving a nearly optimal trade-off between standard risk and adversarial risk. However, this point seems rather obvious.
>
> The mentioned statement is merely there to emphasize how requiring a robust predictor/estimator to achieve nearly optimal (standard) risk translates into a constraint on ϵ. Quantifying the threshold on ϵ is problem specific; the example in Section 3 provides a template for such an analysis.
>
> > Overall, I feel that this paper offers a valuable theoretical examination of an existing intuition, improving over already existing theoretical analyses which use specific assumptions. This study provides lower bounds for the perturbation size, which would still allow for adversarial robustness, however does not sufficiently demonstrate the influence of this finding to commonly used adversarial attacks under ℓ2 and ℓ∞ perturbations in e.g., image classification scenarios. Providing further information on this aspect and its practical applications would enhance the significance of this study. Thus, I would appreciate some small experimental analysis.
>
> Our results are meant to characterize the impossibility simultaneously low adversarial and standard risk if ϵ is greater than a certain threshold. Given the nature of the result, we appreciate it if the reviewer can clarify what kind of experiments would be considered a validation of our result. Regarding practicality, similar to the regression example of Section 3, the ℓ2 and ℓ∞ perturbation can be analyzed for a simple model such as binary classification for a very special data distribution such as the Gaussian mixture model. However, these special cases might not significantly improve our understanding beyond the similar analyses done in the literature. Furthermore, we believe that connecting theory and practice of machine learning (as in the case of adversarial robustness) is mainly restricted by the lack of tractable mathematical models for real-world data (e.g., image classification data). For example, suppose that we show in a rigorous and precise mathematical way that the trade-off between the adversarial and standard risk is completely determined by a certain attribute of the data distribution (e.g. its Poincare constant). But for real-world data we usually don’t have an efficient mechanism to quantify the crucial attribute (say the Poincare constant) even approximately.

---

### Review · Reviewer_75M4 · 2024-12-12

**Summary Of Contributions:**

This paper explores the inherent trade-off between standard and adversarial risks in machine learning models, particularly in regression and classification tasks. The authors formalize the intuition that achieving adversarial robustness often comes at the cost of accuracy, especially when the optimal predictor is not smooth. They provide theoretical foundations for this trade-off and evaluate it in the context of polynomial ridge functions under mild regularity conditions.

**Audience:**

Yes

**Broader Impact Concerns:**

No concern

**Claims And Evidence:**

Yes

**Requested Changes:**

- empirical validation and experiments of the theoretical results are needed
- more discussions how the theoretical results can broadly benefit real-world machine learning applications

**Strengths And Weaknesses:**

**Strength**
- The paper provides a solid theoretical basis for understanding the trade-off between accuracy and robustness. The use of mathematical formalism to derive these trade-offs is rigorous and well-justified.
- The results are presented in a general framework that can be adapted to various types of perturbation constraints, making the findings broadly applicable across different machine learning tasks.
- The paper includes concrete examples, such as least-squares regression and multiclass classification, to illustrate the theoretical results. This helps in understanding the practical implications of the trade-offs.

**Weakness**
- I like the theoretical results but I think empirical validation will be essential to demonstrate the practical impact of the proposed trade-offs would strengthen the paper.
- The discussion on the practical implications of the trade-offs is limited. It will greatly enhance the applicability of the theoretical results if there are more discussions on examples of conditions (2) and (3), and how these trade-offs can be managed in real-world applications.

---

> ### Author Response · Authors · 2025-02-12
>
> We thank the reviewer for his/her comments. Below is our response to your comments.
>
> > I like the theoretical results but I think empirical validation will be essential to demonstrate the practical impact of the proposed trade-offs would strengthen the paper.
>
> We appreciate it if the reviewer can clarify what type of empirical validation or experiments the reviewer is considering. Our results intended to show, in a general setting, the impossibility of simultaneously maintaining nearly optimal adversarial robustness and prediction/estimation accuracy if ϵ is relatively large.
>
> > The discussion on the practical implications of the trade-offs is limited. It will greatly enhance the applicability of the theoretical results if there are more discussions on examples of conditions (2) and (3), and how these trade-offs can be managed in real-world applications.
>
> We have added two subsections at the end of the introduction to elaborate on the main contributions, remarks, and future direction. It is possible to formulate a certain mathematical attribute of the data distribution that determines a threshold for ϵ beyond which robustness and having nearly optimal standard risk become incompatible. As long as we can approximate the mentioned attribute, perhaps using other quantifiable attributes of the data distribution, we can use our approach and find the desired threshold for ϵ. It is virtually impossible to make any informative statement if the real-world data lacks any formal mathematical characterization.
>
> The equations (2) and (3) are abstraction of the fact that the commonly used loss functions satisfy certain "quasi triangle inequalities" that become important as they allow as to extract the sharpness of the test function from the adversarial risk.

---

### Review · Reviewer_4LiX · 2025-02-02

**Summary Of Contributions:**

This paper established a lower bound result to connect the adversarial robustness and clean performance. Under some smoothness conditions, the sum of adversarial robustness and clean performance is lower bounded by some terms related to the best model performance and attack strength. Multiple examples of data distributions are provided to illustrate how to use the theory.

**Audience:**

Yes

**Claims And Evidence:**

Yes

**Requested Changes:**

Please address my concerns in the weakness section.

**Strengths And Weaknesses:**

(1) The main weakness is the limited impact and contribution: From literature such as

Javanmard, Adel, Mahdi Soltanolkotabi, and Hamed Hassani. "Precise tradeoffs in adversarial training for linear regression." Conference on Learning Theory. PMLR, 2020.

Javanmard, Adel, and Mahdi Soltanolkotabi. "Precise statistical analysis of classification accuracies for adversarial training." The Annals of Statistics 50.4 (2022): 2127-2156.

The existence of the trade-off of adversarial robustness and clean performance is a known result. From the current presentation of the submission, it only supplements with more data distribution and loss functions for the trade-off, and there is limited justification on why the new results are more important than existing results.

(2) The authors need to provide discussions on the implication of the new results. For example, how are the new results related to the practical adversarial training tasks in image classification, e.g., CIFAR-10? Existing literature attempts to enhance the adversarial robustness of such a task, and can achieve a (clean acc, robust acc) of (>90%, >70%). Can your theoretical results deliver some practically useful conjectures of the upper limit of the performance?

(3) The authors may alternatively justify the technical contribution of this paper. From how the current results are presented, there is no highlight on any technical challenge in deriving the bounds.

---

> ### Author Response · Authors · 2025-02-12
>
> We thank the reviewer for his/her comments. Below are our response to your comments.
> > (1) The main weakness is the limited impact and contribution: From literature such as
>
> >Javanmard, Adel, Mahdi Soltanolkotabi, and Hamed Hassani. "Precise tradeoffs in adversarial training for linear regression." Conference on Learning Theory. PMLR, 2020.
>
> >Javanmard, Adel, and Mahdi Soltanolkotabi. "Precise statistical analysis of classification accuracies for adversarial training." The Annals of Statistics 50.4 (2022): 2127-2156.
>
> >The existence of the trade-off of adversarial robustness and clean performance is a known result. From the current presentation of the submission, it only supplements with more data distribution and loss functions for the trade-off, and there is limited justification on why the new results are more important than existing results.
>
> Our intention has been to express the trade-off between the standard and adversarial risk of function in terms of its “sharpness” in a formal mathematical formulation that applies broadly and under minimal assumptions. By focusing on problems with special statistical models and data distributions, prior results provide more details on the nature of the trade-off as it occurs in those problems. However, we also need to understand what causes the trade-off to occur in a generic setting, and be cautious about generalizing the phenomena observed under the special models. For example, most of the models for which adversarial robustness is studied mathematically are basically linear models. In classification, for instance, the trade-off between accuracy and robust accuracy may manifest itself completely differently if the data is concentrated around nested spherical shells, or in a multidimensional checkerboard pattern, or on the intricate level sets of a large neural network.
>
> > (2) The authors need to provide discussions on the implication of the new results. For example, how are the new results related to the practical adversarial training tasks in image classification, e.g., CIFAR-10? Existing literature attempts to enhance the adversarial robustness of such a task, and can achieve a (clean acc, robust acc) of (>90%, >70%). Can your theoretical results deliver some practically useful conjectures of the upper limit of the performance?
>
> Without having a mathematically tractable and accurate model for real-world data (e.g., CIFAR-10), it is not possible to provide a precise statement on adversarial robustness (or other aspects) of a learning procedure. For example, the analysis of Section 3 can be generalized to derive a threshold for ϵ beyond which adversarial risk is significantly larger than the optimal standard risk. One critical factor in this threshold is a quantity that resembles the Poincaré constant of the data distribution. So we know the attributes of the data distribution that influence the adversarial robustness, but estimating these attributes for an arbitrary data distribution may only be possible through direct evaluation.
>
> > (3) The authors may alternatively justify the technical contribution of this paper. From how the current results are presented, there is no highlight on any technical challenge in deriving the bounds.
>
> We did not intend to have a technically challenging analysis. Rather we aimed at exposing the properties that are central to the occurrence of the trade-off between standard and adversarial risk. Nevertheless, while we do not consider  the calculations carried out for the example in Section 3 as challenging, they are not entirely trivial either.

---

### Decision · Action_Editor_WQwJ · 2025-03-10

**Recommendation:** Reject

**Comment:**

During the author-reviewer discussions, all the reviewers unanimously have a concern about the relevance of this paper to the community's interest. This is mainly because the presented result (Theorem 1) is rather overly abstract, compared with the previous theoretical studies such as Javanmard et al. In response, the authors argue that the main benefit brought by this paper is a general result removing restricted data/model assumptions.

The AC understands the both standpoints. Indeed, the previous theoretical studies in adversarial robustness mostly assume either the model linearity or Gaussian data assumption, which is far from the practical relevance. The authors characterize the fundamental lower bound of the adversarial risk by the two terms---the model local sharpness and the smoothness factor of the conditional distribution Y|X, without relying on specific structural assumptions. Furthermore, this result is instantiated for the least-square problem in Section 3.

This is great---however, as it remains elusive how we can benefit from this result. Qualitatively, many previous papers have mentioned that the model smoothness (the opposite of the sharpness) matters in adversarial robustness (e.g., the Parseval net https://arxiv.org/abs/1704.08847, just to mention a few). In this context, we would expect the theory to have a "predictive power" to some extent. The AC does believe we can expect such a predictive power for the authors' result. For example, the lower bound in eq. (6) seems to be possible to estimate (at least approximately). This is what some reviewers request for the additional experiments. The additional experiments in this context do not aim to boost our understanding to adversarial robustness but rather verify how predictive the theory is. We could begin from some synthetic datasets (for which we have a tractable conditional distribution Y|X) to validate the tightness of the lower bound, and moreover, estimate the conditional distribution to approximate the lower bound (6) for more complicated datasets like CIFAR-10. Such experiments may pave a road toward a new method for improving adversarial robustness (though the AC might not expect to develop a new method in this work, given its theoretical nature).

For these reasons, the AC cannot accept this paper for this round.

**Audience:**

Despite that the adversarial robustness is one of the core interests in the machine learning community, the reviewers have a concern that the theoretical results provided in this paper is rather too abstract and may not appeal to the vast majority of the machine learning community.

**Claims And Evidence:**

All of the theoretical claims including the problem setup are stated in a self-contained manner.

**Resubmission Of Major Revision:**

The authors may consider submitting a major revision at a later time.